# Effect of Dietary *L*-Methionine Supplementation on Growth Performance, Carcass Traits, and Plasma Parameters of Starter Pekin Ducks at Different Dietary Energy Levels

**DOI:** 10.3390/ani11010144

**Published:** 2021-01-11

**Authors:** Yongbao Wu, Jing Tang, Junting Cao, Bo Zhang, Ying Chen, Ming Xie, Zhengkui Zhou, Shuisheng Hou

**Affiliations:** State Key Laboratory of Animal Nutrition, Key Laboratory of Animal (Poultry) Genetics Breeding and Reproduction, Ministry of Agriculture and Rural Affairs, Institute of Animal Sciences, Chinese Academy of Agricultural Sciences, Beijing 100193, China; 82101171180@caas.cn (Y.W.); tangjing198601@163.com (J.T.); fightingcaoting@163.com (J.C.); 82101172375@caas.cn (B.Z.); chenying1482@163.com (Y.C.); caasxm@163.com (M.X.); zhouzhengkui@caas.cn (Z.Z.)

**Keywords:** Pekin duck, metabolizable energy, *L*-methionine, growth performance, methionine requirement

## Abstract

**Simple Summary:**

Tremendous improvements have been made in the duck commercial industry, especially in aspects such as the increase of body weight and shortening of the production cycle. Meat duck production has been increasing annually worldwide. Improvements in precise nutrition were crucial for ducks to improve growth efficiency and reduce diet costs. Currently, *L*-Met, a new methionine (Met) source, has been commercially available for duck diet formulation. The bioavailability of *L*-Met is greater than *DL*-Met for growth performance in duck. The present study estimated the Met requirement of starter Pekin ducks from 1 to 21 days of age by supplementing crystal *L*-Met to formulate the diets at different ME levels. The results suggested that the Met requirement of starter Pekin ducks was affected by dietary ME levels. The data potentially provide theoretical support for the utilization of crystalline *L*-Met in duck production.

**Abstract:**

A 2 × 6 factorial experiment was conducted to determine the influences of dietary metabolizable energy (ME) and methionine (Met) levels on growth performance, carcass traits, and plasma biochemical parameters of starter Pekin ducks from 1 to 21 days of age. A total of 600 one-day-old male Pekin ducklings were randomly assigned to 12 groups (six replicates each group and eight ducks per replicate) in a 2 × 6 two-factor arrangement. The basal Met levels of two basal diets (11.54 and 12.52 MJ/kg ME) were 0.31 and 0.29%, respectively. The crystalline *L*-Met was supplemented to yield six diets according to different supplemental levels (0, 0.05, 0.10, 0.15, 0.20, and 0.25%). The results showed that the body weight (BW) and average daily weight gain (ADG) were increased (*p* < 0.05) with increasing dietary Met levels. Dietary ME levels changed from 11.54 to 12.52 MJ/kg increased the BW and ADG (*p* < 0.05) as well as decreased the average daily feed intake and feed to gain ratio (*p* < 0.05). As the dietary Met level increased, leg muscle yield increased (*p* < 0.05). Conversely, increasing the dietary ME level decreased the leg muscle yield (*p* = 0.0024) and increased abdominal fat (*p* < 0.001). Meanwhile, the concentrations of total cholesterol (TCHO), high-density lipoprotein cholesterol (HDLC), and low-density lipoprotein cholesterol (LDLC) in plasma were decreased (*p* < 0.05) when the ME levels of diets changed from 11.54 to 12.52 MJ/kg. Meanwhile, the plasma TCHO and HDLC concentrations decreased (*p* < 0.05) as dietary Met levels increased. Based on the linear-broken line model, the dietary Met requirement of starter Pekin ducks from 1 to 21 days of age for optimal ADG were 0.362% (0.052% supplemental *L*-Met) at 11.54 MJ ME/kg and 0.468% (0.178% supplemental *L*-Met) at 12.52 MJ ME/kg, respectively, when crystal *L*-Met was supplemented to formulate the diets. This suggested that the Met requirement of starter Pekin ducks was affected by dietary ME levels. The data potentially provide theoretical support for the utilization of crystalline *L*-Met in duck production.

## 1. Introduction

Meat duck production has been increasing annually worldwide over the decades [1]. The precise nutrient formulation is crucial to increase the growth of ducks and reduce diet costs and nitrogen emissions. The continued improvements in duck dictated the need for continual re-evaluation of nutrient requirements in diets.

Methionine (Met), usually the first limiting amino acid for ducks [2,3], plays vital roles in protein synthesis, methylation process, and cellular antioxidant capacity [4,5]. Thus, extensive studies on Met requirement and its role in growth of ducks were conducted over past decades [3,6,7]. The crystalline *DL*-Met is usually supplemented to balance the duck Met requirement of ducks. Theoretically, chemically synthesized *DL*-Met supplemented in the diet is a racemic mixture of equivalent *D*- and *L*-Met. *L*-Met could be incorporated directly into the body with approximately 100% bioavailability, but *D*-Met must be converted to *L*-Met before the incorporation into protein [8]. Currently, *L*-Met, a new Met source, has been commercially available for duck diet formulation. The bio-efficacy of *L*-Met was approximately 1.4 times relative to *DL*-Met for the growth performance of starter ducks [9] as well as gut oxidative status and development of chicks [10]. However, to date, few studies have evaluated the Met requirement of starter ducks by dietary *L*-Met supplementation.

Metabolizable energy (ME) is an index to evaluate energy levels in duck diets. High ME levels in broiler diet resulted in a reduction of feed intake [11], which might influence the intake and utilization of other nutrients. Therefore, other nutrient requirements might be altered by dietary ME level. Our previous studies showed that a higher dietary ME level required a greater lysine requirement for starter Pekin ducks [12] and a greater Met requirement for growing Pekin ducks [13]. Thus, we hypothesized that the Met requirement of starter Pekin ducks could be affected by dietary ME levels, which should be further investigated. Therefore, the purpose of this study was to determine the effects of dietary ME and Met levels on growth performance, carcass traits, and plasma biochemical parameters and to estimate the Met requirement of starter Pekin ducks from 1 to 21 days of age.

## 2. Materials and Methods

### 2.1. Experimental Design, Animals, and Housing

All experimental procedures of the present study were permitted by the Animal Care and Use Committee of Institute of Animal Sciences, Chinese Academy of Agricultural Sciences (CAAS) (Approval number: IAS2018-16; approved on 20th April, 2018). A 2 × 6 factorial experiment, using 2 dietary ME levels (11.54, 12.52 MJ/kg) and 6 supplemental Met levels (0, 0.05, 0.10, 0.15, 0.20, and 0.25% *L*-Met), was conducted to evaluate the effects of dietary ME and Met levels on growth performance, carcass characteristics, and plasma biochemical parameters of starter Pekin ducks from 1 to 21 days of age. At 1 day of age, a total of 750 male Pekin ducklings (from the Pekin Duck Breeding Center of CAAS) were weighed individually, and the birds with lowest (below 50 g) and highest (above 70 g) body weight were removed. Finally, a total of 600 male Pekin ducklings (56.1 ± 0.7 g) were selected from the remaining ducklings. Then, these birds were randomly divided into 12 treatments with 6 replicates each and 8 birds per replicate according to similar pen body weight (no significant difference among each pen weight, *p* = 0.71). Each plastic-floor pen (length 2 m × width 0.75 m × height 0.4 m) served as a replicate, and the experiment lasted for 3 weeks (ducklings from 1 to 21 days of age). Pelleted feed and water were available ad libitum, and the water was provided by the drip-nipple water supply line. During the experimental period, the temperature was kept at 32 °C from 1 to 3 days of age and then reduced gradually to approximately 22 °C thereafter until 21 days of age in the duck house. The lighting was kept continuous (24 h) throughout the whole study.

### 2.2. Diets

Two basal diets (11.54, 12.52 MJ/kg ME) were formulated to be Met-deficient with varying percentages of corn, soybean meal, and wheat bran (Table 1). Then, each basal diet was divided into 6 equal sub lots to supplement with crystalline *L*-Met (purity ≥ 99%, CheilJedang Co., Seoul, Korea) according to 6 supplemental levels (0, 0.05, 0.10, 0.15, 0.20, and 0.25%) at the expense of rice hull (Table 1). The formulated diet was pelleted by a flat die pellet mill (2.5 mm). After pelleting, the pellet diet was dried in the shade. The amino acid concentration (including total Met) of two basal diets was determined (L-800, Hitachi, Tokyo, Japan) based on as-fed diets (Table 1). The basal Met levels of 2 basal diets (11.54, 12.52 MJ/kg ME) were 0.31 and 0.29%, respectively (Table 1). The crystalline *L*-Met was supplemented to yield 12 diets containing 0.31, 0.36, 0.41, 0.46, 0.51, and 0.56% total Met levels for 11.54 MJ/kg ME diet, and 0.29, 0.34, 0.39, 0.44, 0.49, and 0.54% total Met levels for 12.52 MJ/kg ME diet, respectively. Except for dietary ME and Met, the other nutrient levels in both basal diets were all met or exceeded the recommendation of Nutrient Requirement of Meat-type Duck in China (NY/T 2122-2012) [14] for starter Pekin ducks.

### 2.3. Growth Performance

At 7, 14, and 21 days of age, all ducks were fasted overnight (10–12 h) for the accuracy and reliability of growth performance; then, the body weight (BW) and diet consumption of the ducks in each pen were measured. Average daily weight gain (ADG), average daily feed intake (ADFI), and feed/gain (F/G) were calculated of birds at different stages (1 to 7 days, 8 to 14 days, 15 to 21 days, and 1 to 21 days). The ADFI and F/G were corrected for mortality. ME and Met intake were determined according to the ME and Met concentrations in diets and ADFI. Meanwhile, the ME and Met conversion were calculated according to the formula [15]: ME conversion = ME intake/ADG, and Met conversion = Met intake/ADG.(1)

### 2.4. Carcass Characteristics

At 21 days of age, all ducks were fasted overnight (10–12 h) for the accuracy and reliability of carcass traits, and two ducks with close to the average body weight were randomly selected from each pen. Then, the selected ducks were weighed and euthanized by CO_2_ inhalation and were immediately bled. The breast muscle, leg muscle (deboned and peeled), and abdominal fat were all separated manually, and their weights were recorded. The yields of breast muscle, leg muscle, and abdominal fat were determined as the following equation:Yield = (breast muscle, leg muscle, or abdominal fat weight) × 100%/individual body weight.(2)

### 2.5. Plasma Biochemical Parameters

At 21 days of age, all ducks were fasted overnight (10–12 h) before blood collection for the accuracy and reliability of plasma biochemical parameters. Before duck euthanizing, the heparinized blood of two ducks selected for carcass traits measuring was collected using heparin sodium-anticoagulant tubes from the wing vein and centrifuged at 1300 × *g* for 15 min at 4 °C to obtain the plasma samples. The plasma samples were stored at −20 °C for further analysis. The plasma concentration of triglyceride (TG), total cholesterol (TCHO), high-density lipoprotein cholesterol (HDLC), low-density lipoprotein cholesterol (LDLC), alanine transaminase (ALT), aspartate transaminase (AST), total protein (TP), albumin (ALB), uric acid (UA), glucose (GLU), and alkaline phosphatase (ALP) were measured based on a spectrophotometric method using an automatic analyzer (Hitachi 7080, Tokyo, Japan) with corresponding kits (Maccura, Chengdu, China) based on their specifications.

### 2.6. Statistical Analysis

The data were analyzed as a 2 × 6 factorial experiment design by the two-way ANOVA with GLM procedure (SAS Institute, Cary, NC, USA). The replicate (pen) was considered to be the experimental unit for growth performance, while each duck was considered to be the unit for carcass traits and plasma biochemical parameters. The main effects of dietary ME level, Met level, and their interaction of each parameter was evaluated. Results were presented as means with the pooled standard error of the means (SEM). In the case of significant treatment (*p* < 0.05) between dietary ME or Met level, Duncan’s multiple comparison tests were used to compare differences among means. Based on main effects, polynomial contrasts were performed to assess the linear and quadratic responses of dietary Met levels on dependent variables. In the present study, the linear broken-line regression analysis [16] was used to estimate the dietary Met requirements of starter Pekin ducks at 2 dietary ME levels using the PROC NLIN procedure (SAS Institute). The linear broken-line model was as follows:y = l + u (x − r), when x ≤ r 
y = l, when x > r 
where y = dependent variable, x = independent variable, r = Met requirement, l = theoretical maximal ADG, and u = rate constant. In addition, the t-test was conducted to compare both Met requirements between two dietary ME levels according to the reported statistical method [17].

## 3. Results

### 3.1. Growth Performance

As is shown in Table 2, the BW at 21 days and ADG from 1 to 21 days of age were increased (*p* < 0.001), while the ADFI (*p* = 0.0035) and F/G (*p* < 0.001) were decreased when dietary ME changed from 11.54 to 12.52 MJ/kg. Dietary *L*-Met supplementation increased the BW at 21 days and ADG from 1 to 21 days of age (*p* < 0.05) in both ME levels diets. The ADFI was linearly increased (*p* < 0.05) as dietary *L*-Met supplementation. Compared with ducks fed 11.54 MJ ME/kg diet, birds fed 12.52 MJ ME/kg of diet had the greater ME intake (*p* = 0.0014), and the lower ME conversion (*p* < 0.001), Met intake (*p* = 0.0023), and Met conversion (*p* < 0.001). The Met intake (*p* < 0.001) and Met conversion (*p* < 0.001) of ducks as increased dietary Met supplementation (Table 3).

The growth performance of starter Pekin ducks during different stages (1 to 7 days, 8 to 14 days, and 15 to 21 days) was investigated (Figure 1 and Figure 2). The BW of ducks was no different (*p* > 0.05) at the beginning of the trial (day 1), while the birds fed 12.52 MJ ME/kg diet had higher (*p* < 0.05) BW at day 14 and day 21. Similarly, the ADG of ducklings from 8 to 14 days and 15 to 21 days of age fed 12.52 MJ ME/kg diet as greater (*p* < 0.05) than birds fed 11.54 MJ ME/kg diet. During the whole study period, dietary ME level drove to reduce the ADFI and F/G (*p* < 0.05) of ducks at each stage (1 to 7 days, 8 to 14 days, and 15 to 21 days of age). As dietary *L*-Met supplementation, the ducks at 7 days, 14 days, and 21 days had all greater BW compared to birds fed diets without *L*-Met supplementation. The ADG and ADFI of ducklings at the first two weeks (1 to 7 days, 8 to 14 days) increased as dietary *L*-Met supplementation increased.

### 3.2. Carcass Characteristics

The effects of dietary ME and Met levels on the carcass characteristics of starter Pekin ducks at 21 days of age are shown in Table 4. The leg muscle yield was reduced (*p* = 0.0024) and the abdominal fat (*p* < 0.001) was increased when the dietary ME level rose from 11.54 to 12.52 MJ/kg. As dietary *L*-Met supplementation increased, the leg muscle yield was increased (*p* < 0.05). In addition, dietary *L*-Met supplementation resulted in a tendency to increase breast muscle yield (*p* = 0.0904) and a decreasing tendency for abdominal fat (*p* = 0.0767) although this was not significant.

### 3.3. Plasma Biochemical Parameters

The concentrations of TCHO (*p* = 0.0002), HDLC (*p* = 0.0005), and LDLC (*p* = 0.0287) in plasma were decreased when the ME levels of diets increased from 11.54 to 12.52 MJ/kg (Table 5). The plasma TCHO and HDLC concentration were decreased (*p* < 0.05) as dietary *L*-Met supplementation increased (Table 5). The plasma TP, ALB, GLU, UA, and ALP concentrations were not affected (*p* > 0.05) by dietary ME or Met levels (Table 6).

### 3.4. Met Requirements

The Met requirements of starter Pekin ducks were estimated by the regression model (Table 7). Based on the linear-broken line model, the dietary Met requirement of starter Pekin ducks from 1 to 21 days of age for optimal ADG were 0.362% (0.052% supplemental *L*-Met) at 11.54 MJ ME/kg, and 0.468% (0.178% supplemental *L*-Met) at 12.52 MJ ME/kg, respectively, when crystal *L*-Met was supplemented to formulate the diets. The dietary Met requirement of 0.468% at 12.52 MJ ME/kg was greater (*p* < 0.05) than 0.362% at 11.54 MJ ME/kg by Student *t*-test analysis.

## 4. Discussion

In China, the corn, soybean meal, and wheat bran are the most common feed ingredients to formulate the poultry diets. Therefore, the corn-soybean based diet was used in the present study, in which Met was the first limiting amino acid for poultry. The wheat bran or soybean oil was partly replaced (about 3–4%) by the corn and soybean meal to formulate the low- or high-energy basal diet. Both basal diets were Met-deficient (0.31% and 0.29% Met, analyzed) for starter Pekin ducks to investigate the growth responses. In our study, growth depression was observed when ducklings fed both Met-deficient basal diets, and BW and ADG of birds were improved after feeding diet supplemented crystalline *L*-Met (Table 1). Similarly, previous studies showed that Met deficiency resulted in a decrease in BW and ADG as well as an increase in F/G of broiler [18,19], duck [3], and goose [20], which was counteracted via the dietary Met supplementation. Energy is crucial for growth in the diet of ducks, and ME is a common criterion of dietary energy utilization response in duck diets, which influences the feed intake [11]. As expected, in the current experiment, the BW and ADG were increased, while ADFI and F/G were decreased when dietary ME increased from 11.54 to 12.52 MJ/kg. The results were in line with our previous studies, which showed that an increase in ADG and a decrease in ADFI and F/G as dietary increased ME levels in the diet of starter [21] and growing [13,22] Pekin ducks. In addition, as the dietary ME level decreased from 12.52 to 11.54 MJ/kg, we observed that the BW and ADG were significantly decreased starting at the second week (Figure 1), which might be due to the yolk sac in the body of the duckling that provided the nutrients for the first week [23,24]. However, Met deficiency led to poor growth performance starting in the first week (Figure 2), which suggested that the growth response to dietary Met concentration for ducklings of one week of age might be greater than that to dietary ME concentration. Furthermore, the ADG and ADFI of ducks fed the Met-deficient diet have no differences compared with other ducks fed the diet with Met supplementation at the third week of the feeding period (Figure 2), which implied that the 1-week-old ducklings were more sensitive to dietary Met concentration than birds from 3 weeks of age.

Carcass characteristics have been considered as crucial response indicators to evaluate dietary energy and amino acid status in poultry diets such as yields of breast muscle, leg muscle, and abdominal fat [3,13,22,25]. Despite the ducks not yet having reached market weight, the carcass characteristics of birds at 21 days of age that were measured partly represented the status of muscle and fat production. In the current study, the abdominal fat of ducks fed a 12.52 MJ/kg ME diet was increased compared to birds in the low-energy group (Table 4). A similar phenomenon was also found in the previous starter broiler [26] and starter gosling [27] studies indicating that increasing dietary energy caused an excessive accumulation of abdominal or visceral fat. Several studies showed that no differences were observed on effects of high-energy diets on the yield of breast muscle of broilers [11,28,29] or Pekin ducks [22], which is consistent with the results of the present study (Table 4). However, the result of leg muscle yield (Table 4) was contradictory to previous studies [22]. It is known that Met is an essential amino acid in duck diet, the status of which affected the weight gain and carcass composition of ducks, and Met deficiency could lead to a decrease in edible carcass components and an increase in non-edible offal [13]. In the present study, the leg muscle yield of duck at 21 days of age was increased when ducks were fed dietary Met-supplemented diets. In addition, there was a tendency to increase breast muscle yield and a tendency to decrease the abdominal fat (Table 4). These results were in accordance with our previous studies [3,13]. Ingredients supplying dietary energy and crude protein (CP) represent most of the diet cost for meat ducks. A balanced ME/CP ratio was required to achieve optimum growth performance and carcass traits for Pekin ducks [15,30]. Generally, a diet high in CP (or amino acids, AAs) yields more meat and less fat deposition [13,15], which benefits Pekin duck production by increasing edible carcass components (breast and leg muscle) and reducing non-edible offal (abdominal fat). However, the costs were too expensive if diets were formulated with high CP or AAs levels. In addition, the amounts of protein in the hindgut were increased if animals had dietary high protein or amino acids levels in their diet. The protein in the hindgut could be fermented by microorganisms, which might promote the colonization of pathogenic bacteria in the gut.

Plasma or serum biochemical parameters are reflection of the nutrient metabolism condition in ducks. Plasma TCHO mainly comes from diets or was synthesized in the body. Due to the similar proportion of formula for both basal diets (without animal protein source), the difference of plasma TCHO concentration observed in the present study (Table 5) was a result of the endogenous synthesis of TCHO by ducks. The key function of HDLC, which is referred to as good cholesterol, is the transportation of TCHO and fat from other tissue cells to the liver [31]. The plasma TCHO and HDLC concentrations were both decreased as dietary ME and Met levels increased (Table 5). The decreased HDLC might be due to the reduction of TCHO concentration in plasma. The earlier studies in humans have shown that the serum HDLC concentration was negatively associated with the BW [32,33]. In our study, the BW of ducks was increased, whereas the plasma HDLC concentration was decreased when the dietary ME or Met level increased, which was consistent with the aforementioned studies in human [32,33]. Plasma UA, the end product of amino acid degradation, could be selected as an indicator of amino acid utilization [34,35]. The plasma TP and ALB concentrations might reflect the protein metabolism status. In our study, no differences were observed for TP and ALB among the groups (Table 6), which was in agreement with the previous study in broiler chicken [36]. This suggested that amino acid metabolism might not be susceptible to dietary ME or Met levels in starter ducklings.

Several regression models, such as quadratic polynomial [3,7,37], linear [12,13], or quadratic broken-line [38,39] models, were fitted to estimate the amino acids requirements for poultry previously, and each model may have advantages and disadvantages [40]. Taken together, linear broken-line regression was used to estimate the Met requirement for optimal ADG (a vital indicator of growth performance) of starter ducklings in our study. The results showed that the dietary Met requirement was different (0.362% versus 0.468%) at both ME level diets (Table 7). This phenomenon agreed well with our recent studies of lysine requirement for starter Pekin ducks [12] and the Met requirement of growing Pekin ducks [13]. Birds have the ability to adjust the feed intake according to the energy level in diets, and energy level is usually selected as the basis for setting other nutrient concentrations in formulating diets of poultry [41]. Thus, the greater Met requirement was needed due to the lower feed intake when ducks were fed the high-energy diets. Our earlier study showed that the optimal dietary Met requirement of Pekin ducklings from hatch to 21 days of age for ADG was 0.481% in the diet of 12.3 MJ/kg ME diet [6]. The results of the Met requirement in the present study were lower compared with 0.481%, the explanation of which may be due to different dietary ME levels or genetic lines of birds. In addition, another vital factor might be that crystal *L*-Met, not *DL*-Met, was supplemented to formulate the experimental diets in the current study. The bio-efficacy of *L*-Met to *DL*-Met ranged from 120 to 140% for the growth performance of young ducks from 1 to 14 days of age [9]. Similarly, our previous study also showed that the bio-efficacies of *L*-Met relative to *DL*-Met were 134% of Pekin ducks from 1 to 21 days of age by the linear ratio-slope regression [10]. That is to say, the utilization of *L*-Met for starter ducks was 1.2 to 1.4 times that of *DL*-Met, which might be the main factor resulting in the lower Met requirement in our study. Considering the utilization efficiency and costs of crystal *L*-Met, the supplementation of *L*-Met in duck diet might be reasonable and economic. Therefore, the results provided the data for the application of crystalline *L*-Met in the duck diet.

## 5. Conclusions

In conclusion, the growth performance, carcass traits, and plasma biochemical parameters could be regulated by dietary ME or Met levels in starter ducks. Based on the linear broken regression model, the dietary Met requirement of starter Pekin ducks from 1 to 21 days of age for ADG was 0.362% (0.052% supplemental *L*-Met) at 11.54 MJ ME/kg and 0.468% (0.178% supplemental *L*-Met) at 12.52 MJ ME/kg, respectively, when crystal *L*-Met was supplemented to formulate the corn–soybean meal type diets. This suggested that the Met requirement of starter Pekin ducks was affected by dietary ME levels. The data potentially provide theoretical support for the utilization of crystalline *L*-Met in duck production.

## Figures and Tables

**Figure 1 animals-11-00144-f001:**
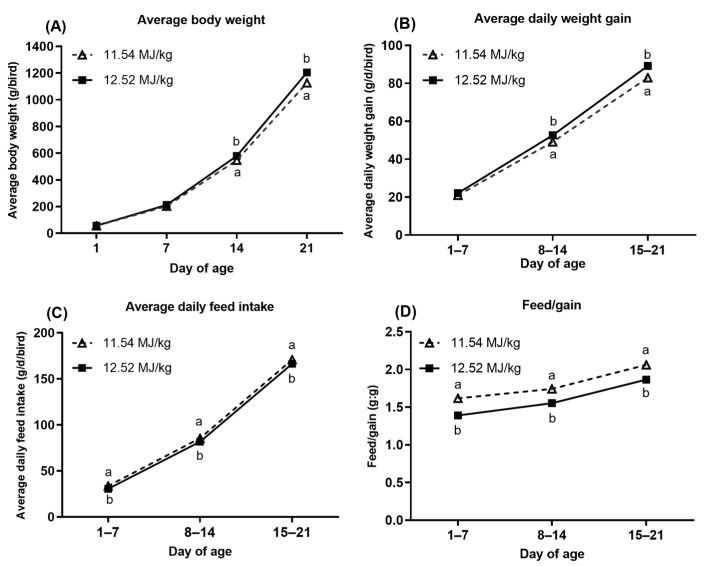
Effect of dietary energy concentration on growth performance of starter Pekin ducks (1 to 7, 8 to 14, 15 to 21 days of age). (**A**): Average body weight; (**B**): average daily weight gain; (**C**): average daily feed intake; (**D**): feed/gain. ^a,b^ Means within the same day of age with different superscripts are significantly different (*p* < 0.05). The results were obtained from data of main effect (metabolizable energy, ME) by two-way ANOVA analysis. SEM = 0.12, 3.23, 5.40, and 7.52 for 1, 7, 14, and 21 day body weight, SEM = 0.46, 0.45, and 0.51 for average daily weight gain of 1 to 7, 8 to 14, and 15 to 21 days of age, SEM = 0.56, 0.88, and 1.46 for average daily feed intake of 1 to 7, 8 to 14, and 15 to 21 days of age, SEM = 0.015, 0.010, and 0.012 for feed to gain ratio of 1 to 7, 8 to 14, and 15 to 21 days of age.

**Figure 2 animals-11-00144-f002:**
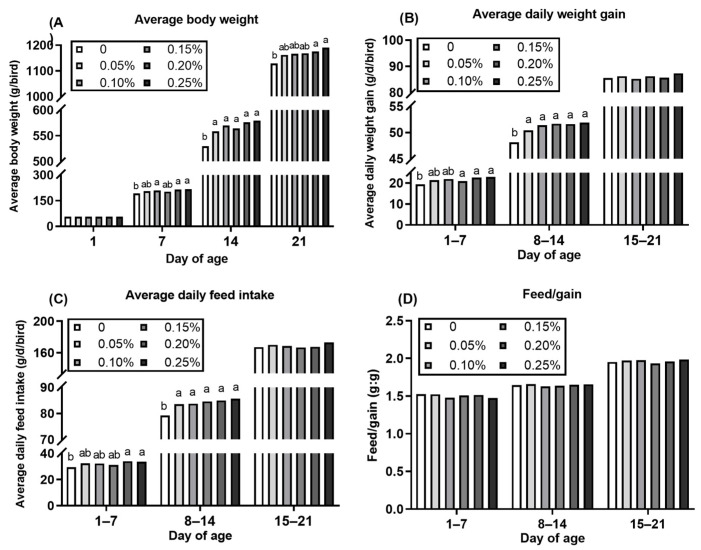
Effect of dietary methionine concentration on growth performance of starter Pekin ducks (1 to 7, 8 to 14, and 15 to 21 days of age). (**A**): Average body weight; (**B**): average daily weight gain; (**C**): average daily feed intake; (**D**): feed/gain. ^a,b^ Means within the same day of age with different superscripts are significantly different (*p* < 0.05). The results were obtained from data of main effect (methionine, Met) by two-way ANOVA analysis. SEM = 0.21, 5.60, 9.36, and 13.02 for 1, 7, 14, and 21 day body weight, SEM = 0.80, 0.78, and 0.88 for average daily weight gain of 1 to 7, 8 to 14, and 15 to 21 days of age, SEM = 0.98, 1.53, and 2.54 for average daily feed intake of 1 to 7, 8 to 14, and 15 to 21 days of age, SEM = 0.026, 0.017, and 0.021 for feed to gain ratio of 1 to 7, 8 to 14, and 15 to 21 days of age.

**Table 1 animals-11-00144-t001:** Composition and nutrient levels of basal diets (as-fed basis, % unless otherwise stated).

Item	Dietary ME Level/(MJ/kg)	
	11.54	12.52
Ingredients		
Corn (8.1% CP)	60.25	58.00
Soybean meal (43.6% CP)	32.23	33.74
Soybean oil	-	3.71
Wheat bran (16.5% CP)	3.00	-
Dicalcium phosphate	1.77	1.80
Limestone	1.20	1.20
Salt	0.30	0.30
Rice hull + *L*-Met ^1^	0.25	0.25
Premix ^2^	1.00	1.00
Total	100.00	100.00
Nutrient levels		
ME/(MJ/kg) ^3^	11.54	12.52
Crude protein ^4^	19.96	20.09
Lysine ^4^	0.93	1.01
Methionine ^4^	0.31	0.29
Methionine + cystine ^4^	0.55	0.55
Tryptophan ^3^	0.23	0.24
Arginine ^4^	1.16	1.24
Threonine ^4^	0.69	0.71
Valine ^4^	0.88	0.91
Isoleucine ^4^	0.70	0.74
Leucine ^4^	1.53	1.57
Calcium ^3^	0.91	0.92
Available phosphorus ^3^	0.40	0.40

^1^ Methionine was supplemented to the diets at the expense of rice hull. ^2^ Supplied per kilogram of total diet: Cu (CuSO_4_·5H_2_O), 8 mg; Fe (FeSO_4_·7H_2_O), 60 mg; Zn (ZnO), 60 mg; Mn (MnSO4·H_2_O), 100 mg; Se (Na_2_SeO_3_), 0.3 mg; I (KI), 0.4 mg; choline chloride, 1000 mg; vitamin A (retinyl acetate), 4000 IU; vitamin D_3_ (cholcalciferol), 2000 IU; vitamin E (dl-α-tocopheryl acetate), 20 IU; vitamin K_3_ (menadione sodium bisulfate), 2 mg; thiamin (thiamin mononitrate), 2 mg; riboflavin, 10 mg; pyridoxine hydrochloride, 4 mg; cobalamin, 0.02 mg; calcium-d-pantothenate, 20 mg; nicotinic acid, 50 mg; folic acid, 1 mg; and biotin, 0.15 mg. ^3^ Values were calculated according to the formulation. ^4^ Values were determined based on triplicate analysis.

**Table 2 animals-11-00144-t002:** Effects of dietary metabolizable energy and methionine levels on growth performance of starter Pekin ducks from 1 to 21 days of age ^1,2^.

ME Level (MJ/kg)	Met Supplemental Level (%)	BW (g/bird)	ADG (g/bird/day)	ADFI (g/bird/day)	F/G (g/g)
11.54	0	1091 ^f^	49.3 ^f^	92.3 ^b^	1.89 ^a,b^
	0.05	1138 ^d,e,f^	51.5 ^d,e,f^	97.9 ^a,b^	1.90 ^a,b^
	0.10	1132 ^d,e,f^	51.2 ^d,e,f^	97.4 ^a,b^	1.90 ^a,b^
	0.15	1119 ^e,f^	50.6 ^e,f^	93.3 ^b^	1.87 ^b^
	0.20	1129 ^d,e,f^	51.1 ^d,e,f^	97.0 ^a,b^	1.90 ^a,b^
	0.25	1153 ^c,d,e^	52.2 ^c,d,e^	101.0 ^a^	1.93 ^a^
12.52	0	1168 ^b,c,d,e^	52.9 ^b,c,d,e^	91.0 ^b^	1.72 ^c^
	0.05	1187 ^a,b,c,d^	53.8 ^a,b,c,d^	92.1 ^b^	1.72 ^c^
	0.10	1202 ^a,b,c^	54.5 ^a,b,c^	92.2 ^b^	1.69 ^c^
	0.15	1217 ^a,b^	55.3 ^a,b^	93.9 ^b^	1.70 ^c^
	0.20	1223 ^a,b^	55.6 ^a,b^	93.9 ^b^	1.69 ^c^
	0.25	1229 ^a^	55.9 ^a^	93.9 ^b^	1.68 ^c^
Pooled SEM		18.4	0.87	2.07	0.017
ME (MJ/kg)	11.54	1127 ^b^	51.0 ^b^	96.7 ^a^	1.90 ^a^
	12.52	1204 ^a^	54.7 ^a^	92.8 ^b^	1.70 ^b^
Pooled SEM		7.2	0.34	0.84	0.007
Supplemental Met (%)	0	1129 ^b^	51.1 ^b^	91.7	1.80
	0.05	1162 ^a,b^	52.7 ^a,b^	95.0	1.81
	0.10	1167 ^a,b^	52.9 ^a,b^	94.8	1.80
	0.15	1168 ^a,b^	53.0 ^a,b^	93.6	1.78
	0.20	1176 ^a^	53.3 ^a^	95.5	1.79
	0.25	1191 ^a^	54.0 ^a^	97.4	1.81
Pooled SEM		13.0	0.62	1.46	0.012
Probability	ME	<0.0001	<0.0001	0.0035	<0.0001
	Met	0.0416	0.0396	0.1389	0.5965
	ME × Met	0.8030	0.8000	0.4182	0.1117
Met linear response		0.0174	0.0168	0.0288	0.8690
Met quadratic response		0.6293	0.6244	0.9258	0.6453

BW, body weight; ADG, average daily weight gain; ADFI, average daily feed intake; F/G, feed to gain ratio. ^1^ Data are the means of six replicates of eight ducks each. ^2,a–f^ Means with different superscripts differ at *p* < 0.05.

**Table 3 animals-11-00144-t003:** Effects of dietary metabolizable energy and methionine on metabolizable energy and methionine intake and conversion of starter Pekin ducks from 1 to 21 days of age ^1,2^.

ME Level (MJ/kg)	Met Supplemental Level (%)	Met Intake (mg/bird/day)	Met Conversion (mg/g) ^3^	ME Intake (kJ/bird/day)	ME Conversion (kJ/g) ^3^
11.54	0	277 ^g^	5.62 ^h^	1066 ^c^	21.7 ^a,b,c^
	0.05	343 ^f^	6.65 ^f^	1131 ^a,b,c^	21.9 ^a,b^
	0.10	389 ^e^	7.60 ^e^	1125 ^a,b,c^	21.9 ^a,b^
	0.15	420 ^d^	8.29 ^d^	1077 ^b,c^	21.3 ^b,c^
	0.20	485 ^c^	9.49 ^b^	1120 ^a,b,c^	21.9 ^a,b^
	0.25	556 ^a^	10.64 ^a^	1167 ^a^	22.3 ^a^
12.52	0	273 ^g^	5.16 ^i^	1139 ^a,b,c^	21.5 ^b,c^
	0.05	323 ^f^	5.99 ^g^	1154 ^a,b^	21.4 ^b,c^
	0.10	369 ^e^	6.76 ^f^	1155 ^a,b^	21.2 ^c^
	0.15	423 ^d^	7.64 ^e^	1176 ^a^	21.3 ^b,c^
	0.20	470 ^c^	8.45 ^d^	1176 ^a^	21.2 ^c^
	0.25	516 ^b^	9.24 ^c^	1175 ^a^	21.0 ^c^
Pooled SEM		8.7	0.08	24.9	0.23
ME (MJ/kg)	11.54	412 ^a^	8.05 ^a^	1114 ^b^	21.8 ^a^
	12.52	395 ^b^	7.21 ^b^	1162 ^a^	21.2 ^b^
Pooled SEM		3.6	0.03	10.2	0.09
Supplemental Met (%)	0	275 ^f^	5.39 ^f^	1103	21.6
	0.05	333 ^e^	6.32 ^e^	1142	21.7
	0.10	379 ^d^	7.18 ^d^	1140	21.6
	0.15	421 ^c^	7.96 ^c^	1127	21.3
	0.20	477 ^b^	8.97 ^b^	1148	21.5
	0.25	536 ^a^	9.94 ^a^	1171	21.7
Pooled SEM		6.2	0.06	17.6	0.16
Probability	ME	0.0023	<0.0001	0.0014	<0.0001
	Met	<0.0001	<0.0001	0.1584	0.4670
	ME × Met	0.2259	<0.0001	0.4669	0.0615
Met linear response		<0.0001	<0.0001	0.0324	0.8961
Met quadratic response		0.5661	0.4886	0.9437	0.2884

^1^ Data are the means of six replicates of eight ducks each. ^2,a–i^ Means with different superscripts differ at *p* < 0.05. ^3^ ME (Met) conversion = ME (Met) intake/ADG.

**Table 4 animals-11-00144-t004:** Effects of dietary metabolizable energy and methionine levels on carcass characteristics of starter Pekin ducks at 21 days of age ^1,2^.

ME Level (MJ/kg)	Met Supplemental Level (%)	Breast Muscle (%)	Leg Muscle (%)	Abdominal Fat (%)
11.54	0	1.93	9.83 ^a,b,c,d^	0.81 ^a,b,c^
	0.05	1.88	9.54 ^c,d^	0.87 ^a,b,c^
	0.10	2.07	10.23 ^a,b,c^	0.73 ^a,b^
	0.15	2.17	10.59 ^a^	0.66 ^a,b,c^
	0.20	2.11	10.08 ^a,b,c,d^	0.79 ^a^
	0.25	2.23	10.47 ^a,b^	0.81 ^a,b^
12.52	0	2.02	9.59 ^c,d^	0.96 ^a,b,c,d^
	0.05	2.13	9.35 ^d^	0.98 ^b,c,d^
	0.10	1.92	9.56 ^c,d^	0.84 ^d^
	0.15	2.10	9.82 ^a,b,c,d^	0.94 ^c,d^
	0.20	2.20	9.70 ^b,c,d^	0.86 ^a,b,c^
	0.25	2.17	10.06 ^a,b,c,d^	0.86 ^a,b,c,d^
Pooled SEM		0.10	0.26	0.05
ME (MJ/kg)	11.54	2.07	10.12 ^a^	0.78 ^b^
	12.52	2.09	9.68 ^b^	0.91 ^a^
Pooled SEM		0.04	0.10	0.02
Met (%)	0	1.97	9.71 ^a,b^	0.89
	0.05	2.01	9.45 ^b^	0.93
	0.10	2.00	9.89 ^a,b^	0.79
	0.15	2.13	10.20 ^a^	0.80
	0.20	2.16	9.89 ^a,b^	0.82
	0.25	2.20	10.26 ^a^	0.83
Pooled SEM		0.07	0.18	0.04
Probability	ME	0.6503	0.0024	<0.0001
	Met	0.0904	0.0130	0.0767
	ME × Met	0.3433	0.8204	0.3791
Met linear response		0.0038	0.0039	0.0934
Met quadratic response		0.7452	0.8056	0.1813

^1^ Data are the means of six replicates of two ducks each. ^2,a–d^ Means with different superscripts differ at *p* < 0.05.

**Table 5 animals-11-00144-t005:** Effects of dietary metabolizable energy and methionine levels on lipid metabolism-related biochemical parameters of starter Pekin ducks at 21 days of age ^1,2^.

ME Level (MJ/kg)	Met Supplemental Level (%)	TG (mmol/L)	TCHO (mmol/L)	HDLC (mmol/L)	LDLC (mmol/L)
11.54	0	0.79	6.24 ^a^	3.06 ^a^	1.54
	0.05	0.79	5.76 ^a,b,c^	2.82 ^a,b,c,d^	1.66
	0.10	0.79	6.04 ^a,b^	2.83 ^a,b,c,d^	1.73
	0.15	0.76	5.78 ^a,b,c^	2.73 ^b,c,d^	1.76
	0.20	0.84	5.65 ^a,b,c^	2.74 ^b,c,d^	1.71
	0.25	0.75	6.05 ^a,b^	2.98 ^a,b^	1.73
12.52	0	0.75	5.79 ^a,b,c^	2.87 ^a,b,c^	1.43
	0.05	0.76	5.66 ^a,b,c^	2.75 ^b,c,d^	1.58
	0.10	0.74	5.59 ^b,c^	2.63 ^c,d^	1.74
	0.15	0.78	5.26 ^c^	2.57 ^c,d^	1.50
	0.20	0.77	5.28 ^c^	2.62 ^c,d^	1.68
	0.25	0.8	5.46 ^b,c^	2.54 ^d^	1.50
Pooled SEM		0.04	0.18	0.09	0.05
ME	11.54	0.79	5.92 ^a^	2.86 ^a^	1.69 ^a^
	12.52	0.77	5.50 ^b^	2.66 ^b^	1.57 ^b^
Pooled SEM		0.02	0.08	0.037	0.04
Met	0	0.77	6.02 ^a^	2.96 ^a^	1.48
	0.05	0.77	5.71 ^a,b^	2.78 ^a,b^	1.62
	0.10	0.77	5.81 ^a,b^	2.73 ^b^	1.73
	0.15	0.77	5.52 ^b^	2.65 ^b^	1.63
	0.20	0.80	5.47 ^b^	2.68 ^b^	1.70
	0.25	0.77	5.76 ^a,b^	2.76 ^b^	1.62
Pooled SEM		0.03	0.09	0.07	0.07
Probability	ME	0.4481	0.0002	0.0005	0.0287
	Met	0.9640	0.0418	0.0189	0.1381
	ME × Met	0.7549	0.8243	0.5048	0.6806
Met linear response		0.4378	0.0582	0.0010	0.0006
Met quadratic response		0.9323	0.0241	0.0019	0.0846

TG, triglyceride; TCHO, total cholesterol; HDLC, high-density lipoprotein cholesterol; LDLC, low-density lipoprotein cholesterol. ^1^ Data are the means of six replicates of two ducks each. ^2,a–d^ Means with different superscripts differ at *p* < 0.05.

**Table 6 animals-11-00144-t006:** Effects of dietary metabolizable energy and methionine levels on hepatic damage and function-related biochemical parameters of starter Pekin ducks at 21 days of age ^1^.

ME Level (MJ/kg)	Met Supplemental Level (%)	TP (g/L)	ALB (g/L)	GLU (mmol/L)	UA (μmol/L)	ALP (U/L)
11.54	0	35.5	17.7	8.07	165.6	608.0
	0.05	35.0	17.6	8.27	183.3	727.5
	0.10	36.3	17.9	8.13	173.2	649.1
	0.15	34.8	17.2	7.98	178.2	672.1
	0.20	35.6	17.9	7.96	181.3	631.7
	0.25	36.2	17.9	7.96	173.8	716.5
12.52	0	34.2	17.2	8.27	176.7	634.1
	0.05	36.2	17.8	8.17	180.1	653.0
	0.10	35.2	17.3	8.17	167.2	643.9
	0.15	35.6	17.8	8.23	170.4	672.8
	0.20	35.4	17.5	8.24	168.0	645.9
	0.25	34.7	17.2	8.27	155.3	687.0
Pooled SEM		0.6	0.3	0.19	12.0	43.6
ME	11.54	35.6	17.7	8.06	175.9	667.5
	12.52	35.2	17.5	8.22	169.6	656.1
Pooled SEM		0.2	0.1	0.08	4.9	17.8
Met	0	34.9	17.5	8.17	171.1	621.0
	0.05	35.6	17.7	8.22	181.7	690.2
	0.10	35.7	17.6	8.15	170.2	646.5
	0.15	35.2	17.5	8.11	174.3	672.4
	0.20	35.5	17.7	8.10	174.7	638.8
	0.25	35.4	17.6	8.12	164.5	701.8
Pooled SEM		0.4	0.2	0.1	8.5	30.8
Probability	ME	0.3152	0.2016	0.1419	0.3664	0.6519
	Met	0.7793	0.9388	0.9873	0.8181	0.4047
	ME × Met	0.1300	0.2078	0.8732	0.8789	0.8823

TP, total protein; ALB, albumin; UA, uric acid, GLU, glucose; ALP, alkaline phosphatase.^1^ Data are the means of six replicates of two ducks each.

**Table 7 animals-11-00144-t007:** Methionine requirements of starter Pekin ducks from 1 to 21 days of age fed two different ME levels using the linear broken-line model.

ME Level (MJ/kg)	Regression Equation	Estimated ADG ^1^	Met Requirement ^2^	95% CI	R^2^	*p* Value	*T* Test ^3^
11.54	y = 51.2 − 36.0 × (0.362 − x)	51.2 ± 0.25	0.362 ± 0.016 ^a^	0.311–0.412	0.799	0.090	5.45
12.52	y = 55.8 − 15.8 × (0.468 − x)	55.8 ± 0.09	0.468 ± 0.011 ^b^	0.434–0.501	0.992	0.001	

^1^ The values were presented as Means ± SE. ^2^ Student t test was performed to compare the Met requirements between both dietary ME levels. ^3^ T > T_0.05/30_ = 2.042 means there was a significant difference in both Met requirements. ^a,b^ Values within a column followed by the different lowercases were significantly different (*p* < 0.05).

## Data Availability

Data presented are original and not inappropriately selected, manipulated, enhanced, or fabricated.

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
