# Peer review of "Effect of Dietary L-Methionine Supplementation on Growth Performance, Carcass Traits, and Plasma Parameters of Starter Pekin Ducks at Different Dietary Energy Levels"

_animals, 2021, doi:10.3390/ani11010144_

Round 1

Reviewer 1 Report

Overall, the manuscript presents very interesting results regarding nutrient requirements for ducks.

Major concerns/suggestions are:

1) Flowability of the text could be improved by improving English use. I recommend working with a native English speaker to provide input and editing suggestions.

2) From my perspective this manuscript requires further analysis. Specifically, for the nutrient requirement evaluation. I strongly encourage authors to test different models and different response variables to evaluate how the met requirement changes. There is very useful data such as the carcass characteristics and blood biochemistry that was not discussed. Why these variables were not used to evaluate nutrient requirements??? If the authors already spent time and resources in measuring all these variables, I am sure they can be further discussed in the manuscript. This will also add value for future researchers that would like to replicate your model to evaluate poultry nutrient requirements.

3) A confusion problem was evident regarding wording for supplemental met in the diet….I suggest authors state in the abstract that the basal level of met in diets 0.31% and from this value met as L-met was added to yield 0.36 (0.05+), 0.41 (0.10+), 0.46 (0.15+). etc… % in the diet respectively, for each of the ME levels. I hope I am not confusing the authors. My main problem was not knowing if you were talking about L-met inclusion in the diet or met % concentration in the diet as supplemented by L-met.

4) Please specify how ducks were euthanized for sample collection and add a note on the justification on why such a long fasting period was required for this trial protocol.

Please find additional comments in the attached PDF File.

Author Response

Dear Sir/Madam,

      Our manuscript (animals-1011661) has been revised according to your comments and recommendations and we have uploaded our revision. Responses of point-by-point to the reviewers are included as below. The line number is according to our previous original manuscript in Points and the new line number is according to revised manuscript in Responses. Furthermore, the revisions are highlighted in full text by using the "Track Changes" function in Microsoft Word.

Reviewer 2 Report

The manuscript investigates effect of dietary L-methionine supplementation on growth performance, carcass traits and plasma parameters of starter Pekin ducks at different dietary energy levels. Many clear data were obtained. In the investigation, increased Met level increased BW and abdominal fat%, but did not alter breast muscle%. Which tissues/organs increases did affect BW increase? In this sense, nitrogen(/protein) retention is an important parameter rather than blood biochemical parameters.

Overall

  1. Delete “p-value” or “significantly” if they are mentioned in a sentence because they have the same meaning.
  2. Spell out BW, ADG, TCHO, HDLC, and LDCL at the first appearance.

Introduction

  1. Line 61; “Redox” should be replaced with “oxidative,” although the cited paper used this word in the title.

Methods

  1. 1; Show a scientific name of the duck used in the investigation.
  2. Met, ME conversion; Show the units for the calculation, e.g., g/g or mg/g. And also, describe what those meant.

Results

  1. Tables; Add a horizontal line according to the ME levels, ME, Met, and probability for viewability. 

Author Response

(The authors gave the same response as above.)

Round 2

Reviewer 1 Report

The authors did a good job addressing comments.

Just a few more edits:

Taking as reference your cover letter (response to the reviewer) sent to show changes in the manuscript.

Response 12, 14, and 18 should be included in the text. For the particular case of response 18, no need to add too much detail. A brief note will be enough to indicate why the fasting period was required and will prevent other researchers to be prepared to fast birds before taking samples as you indicated.

I apologize if the comments were not clear or were confusing to authors.  The questions should be answered for readers and not to the reviewer. I think you did excellent comments and would add value to your manuscript as well as future researchers that will try to replicate your experimental materials and methods. I strongly suggest adding these details to your manuscript.

Author Response

Dear Sir/Madam,

      I apologize that we did not make your comments clear. I totally agree with you that the questions be answered for readers and not to the reviewer. Therefore, the manuscript (animals-1011661) has been revised according to your comments and recommendations and we have uploaded our revision. According to Response 12, 14, and 18 in Round1 response to reviewer, we have added the changes in the revised manuscript. The revisions are highlighted in full text by using the "Track Changes" function in Microsoft Word.

Reviewer 2 Report

The authors adequately responded to the comments.

Author Response

Dear Sir/Madam,

     Our manuscript (animals-1011661) has been revised according to your comments and recommendations and we have uploaded our revision. The revisions are highlighted in full text by using the "Track Changes" function in Microsoft Word.

     Thank you so much for reviewing our manuscript.